# Use of Integrative, Complementary, and Alternative Medicine in Children with Epilepsy: A Global Scoping Review

**DOI:** 10.3390/children10040713

**Published:** 2023-04-12

**Authors:** Zahra Zhu, Daniela Dluzynski, Nouran Hammad, Deepika Pugalenthi, Sarah A. Walser, Rea Mittal, Debopam Samanta, Melanie L. Brown, Ali A. Asadi-Pooya, Angelina Kakooza-Mwesige, Alberto Spalice, Martina Capponi, Alain Lekoubou, Ashutosh Kumar, Sita Paudel, Paul R. Carney, Gayatra Mainali, Sunil Naik

**Affiliations:** 1College of Medicine, Penn State University, Hershey, PA 17033, USAgmainali@pennstatehealth.psu.edu (G.M.); snaik2@pennstatehealth.psu.edu (S.N.); 2School of Medicine, Jordan University of Science and Technology, Al-Ramtha 3030, Jordan; 3Department of Psychiatry, Johns Hopkins Hospital, Baltimore, MD 21287, USA; 4Section of Child Neurology, Department of Pediatrics, University of Arkansas for Medical Sciences, Little Rock, AR 72205, USA; 5Department of Pediatrics, Johns Hopkins School of Medicine, Baltimore, MD 21205, USA; 6Epilepsy Research Center, Shiraz University of Medical Sciences, Shiraz 71437, Iran; 7Department of Neurology, Thomas Jefferson University, Philadelphia, PA 19107, USA; 8Department of Pediatrics & Child Health, Makerere University College of Health Sciences, Kampala 7062, Uganda; 9Department of Maternal Sciences, University la Sapienza, 00185 Roma, Italy; 10Department of Neurology, Hershey Medical Center, Penn State University, Hershey, PA 17033, USA; 11Department of Public Health Sciences, College of Medicine, Penn State University, Hershey, PA 17033, USA; 12Department of Pediatrics and Neurology, Penn State Health Milton S. Hershey Medical Center, Hershey, PA 17033, USA; 13Pediatric Neurology Division, University of Missouri Health Care, Columbia, MO 65212, USA

**Keywords:** complementary alternative medicine (CAM), integrative medicine, epilepsy, pediatric, children

## Abstract

(1) Background: Epilepsy is one of the most common chronic neurological disorders in childhood. Complementary and alternative medicine (CAM) use is highly prevalent in patients with epilepsy. Despite CAM’s widespread and increasing popularity, its prevalence, forms, perceived benefits, and potential risks in pediatric epilepsy are rarely explored. (2) Methods: We performed a scoping review of the available literature on the use of CAM in pediatric epilepsy. (3) Results: Overall, global cross-sectional studies showed a variable degree of CAM usage among children with epilepsy, ranging from 13 to 44% in prevalence. Popular types of CAMs reported were supplements, cannabis products, aromatherapy, herbal remedies, dietary therapy, massage therapy, and prayer. Families often report that CAM is effective, although there are limited objective measures of this. Potential risks lie in the use of CAM, such as herbal remedies, and/or unregulated, contaminated, or unpurified products. Studies also underscored inadequate patient–physician discussions regarding CAM. (4) Conclusions: A better understanding of this topic would aid clinicians in guiding patients/families on the use of CAM. Further studies on the efficacy of the different types of CAM used, as well as potential side effects and drug interactions are needed.

## 1. Introduction

Globally, epilepsy is among the most common neurological disorders [1], in addition to headache and stroke [2]. Childhood epilepsies are a heterogeneous group of conditions with differing diagnostic criteria, treatment, and prognosis. It is characterized as recurrent, unprovoked seizures, involving sudden changes in behavior, including the loss of consciousness and sensory or motor disturbances. The International League Against Epilepsy states that epilepsy is diagnosed if one of the following three criteria are met: a patient has (1) 2 or more unprovoked seizures occurring greater than 24 h apart; (2) 1 unprovoked (or reflex) seizure and a probability of further seizures similar to the general recurrence risk (at least 60%) after 2 unprovoked seizures, occurring over the next 10 years; or (3) a diagnosis of an epilepsy syndrome [3]. Treatment involves anti-seizure medications (ASM); however, only about two-thirds of patients are seizure-free while on appropriate ASM. Epilepsy can lead to stigmatization and/or discrimination, and it is associated with increased morbidity (e.g., injury) and mortality [4,5,6]. Patients with epilepsy or their families may decide to use CAM for their condition for various purported reasons.

CAM encompasses medical treatments, therapies, practices, and products that are used outside of conventional allopathic medicine. Alternative medicine uses therapies in place of conventional medicine, whereas complementary medicine is used in addition to conventional medicine, without substituting for it [7]. Integrative medicine focuses on treating the whole person and utilizes appropriate therapies (including CAM), healthcare professionals, and other modalities to address the underlying cause of health problems and achieve optimal healing and health [8].

In certain countries, patients with epilepsy may turn to CAM due to the myths, superstitions, and stigma associated with the disease [9]. Patients or families may also turn to CAM as supplementary to conventional medicines (ASMs) due to religious beliefs [10]. Others may be more motivated to use CAM over conventional treatments as they believe that CAM aligns well with their values, beliefs, and lifestyle [11]. Suboptimal seizure control and a lack of access to conventional medical care are other reasons that may motivate patients or their families to seek CAM [12,13].

The use of CAM is very popular in developing countries, with an estimated 80% of the populations in developing nations relying on CAM for their treatment [14]. In industrialized countries, the percentage of users is also high (around 50%) [14]. As the popularity of CAM is high, there is a trend toward the integration of conventional medicine and CAM [15]. While many believe CAM treatments/therapies to be benign, side effects have been reported [13,15,16]. Herbal remedies, for example, have the potential to interact with ASMs. Herbs may also have intrinsic proconvulsant properties, thereby increasing the risk of seizures [16,17]. Therefore, patient–physician discussions on CAM usage are important to facilitate the safe use of CAM.

In this review, we will discuss (1) the prevalence of CAM use, (2) the most common types of CAM reported, (3) the reported benefits and adverse effects of CAM therapies, (4) the predictors and reasons for using CAM, and (5) the recommendations about appropriate patient–physician communication regarding CAM use in children with epilepsy.

## 2. Materials and Methods

In this scoping study, a search of PubMed MEDLINE and CINAHL was performed on 1 March 2023 using the keywords (Figure 1):

“pediatric*[tw] OR children[tw] OR adolescen*[tw]”AND“complementary and alternative Medicine*”[tw] OR CAM[tw] OR “complementary medicine*”[tw] OR “alternative medicine*”[tw] OR “integrative medicine*”[tw] OR “supplementary medicine*”[tw] OR “holistic medicine*”[tw] OR “dietary supplement*”[tw] OR “complementary therap*”[tw] OR “alternative therap*”[tw] OR “traditional medicine*”[tw] OR “CAM therap*”[tw]AND“epilepsy*”[tw]. PubMed MEDLINE generated 193 results and CINAHL generated 201 articles.

Relevant research articles discovered by this search method were selected manually, including systematic reviews, qualitative studies, cross-sectional studies, randomized controlled trials, and other types of articles, and were included for review. We excluded articles not specifically focused on the pediatric epilepsy population and CAM.

Following this review, a further literature review was conducted by our study team. This was based on topics that arose while reviewing the generated articles, such as traditional medicine, acupuncture, herbs, cannabidiol (CBD). Further scoping of related articles by geography was performed based on the knowledge of our team members. We explored studies from Africa, North and South America, the Middle East, and Asia. This was a quality scoping review, and no statistical analysis was performed.

## 3. Results

### 3.1. Literature Review

Overall, 46 articles were discovered by the database search. Of the 46 articles, 11 were cross-sectional studies regarding the prevalence of CAM usage among the pediatric population with epilepsy. Two additional cross-sectional studies were subsequently found through a related articles search while reviewing the articles generated by the database. The locations, prevalence, and types of CAM used are depicted in Table 1. Of the thirteen studies, eight studies involved self-administered surveys to parents/caregivers and five studies involved structured interviews with the parents/caregivers.

#### 3.1.1. Prevalence and Common Types of CAM Usage

The rates and types of CAM used across the world regions vary (as shown in Table 1) by geographic location, differing across countries and even in different states within some countries.

In the United States, the prevalence of CAM use among pediatric patients with epilepsy in Alabama, Minnesota, and Pennsylvania was reported to be 35, 26, and 13%, respectively [13,18,19]. The most commonly used CAMs included supplements; cannabis-related products, such as CBD oil; aromatherapy; homeopathy; and massage therapy (Table 1). Based on the cross-sectional studies, the CAM prevalence in the international studies was reported to range from 13 to 100%, with common types of CAM being prayer, homeopathy, osteopathy, kinesiology, herbal remedies, acupuncture, diet therapy, and multivitamins.

#### 3.1.2. Types of Commonly Used CAM

Natural products (Herbs/vitamins/supplements)

Historically, people with epilepsy have turned to botanical or herbal products in seeking treatment for their epilepsy. Herbal therapies include traditional Chinese medicine, Ayurveda, and other practices utilizing plant materials for treatment purposes. In the United States, some of the most popular selling herbs are gingko, St. John’s wort, ginseng, Kava, Saw palmetto, and valerian root, among others [30].

It is estimated that almost 90% of people with epilepsy residing in developing countries, often do not receive pharmacological treatment for their disease [31]. While herbal medicine is popular in the United States, in other parts of the world, especially in developing countries, patients are also turning to herbal remedies for the treatment of their epilepsy [32]. The factors for the usage of herbal medications are multifactorial, varying from a lack of access to anti-seizure medications to cultural reasons, such as valuing traditional healers [33].

There are numerous herbs that are reported in the literature to have been studied in mouse models for epilepsy. There are limited clinical trial studies on the safety and efficacy of herbal products. In one study reviewing commonly cited herbal treatments for epilepsy, they found that there was a lack of data supporting antiepileptic properties for seven of the nine common herbal products: American hellebore, betony, blue cohosh, mugwort, pipsissewa, scullcap, and valerian [17]. Kava and mistletoe had laboratory data to support the antiepileptic effects; however, they lacked clinical data support. As is the case for most herbal treatments, this lack of safety data and the potential side effects make it difficult to recommend it as epilepsy treatment.

Although herbal medicine is often perceived to be safe, herbs can have proconvulsant properties [15]. Many can alter cytochrome P450 enzymes and P-glycoproteins, leading to altered ASM levels. For example, borage oil is reported to carry pro-seizure effects and gingko biloba induces cytochrome P450, which can reduce the serum levels of drugs, such as phenytoin and valproate [34]. Herbal medicines also pose a risk as they may be contaminated by heavy metals [35]. A study conducted in Boston investigated 70 Ayurvedic herbal medicine products sold in stores and found that 20% contained potentially harmful levels of mercury, lead, and/or arsenic [36]. This raises the concern for potential heavy metal toxicity, which may increase the risk of seizures [37].

1a.Traditional medicine

Traditional Chinese medicine (TCM) has been widely used in the treatment of epilepsy in China and there are many studies investigating its therapeutic role in treating pediatric epilepsy. A systematic review of 169 randomized controlled trials with 19,542 participants investigated the use of oriental herbal medicine in patients with epilepsy. The study concluded that Oriental herbal medicine as an adjunctive or alternative therapy to ASDs showed promising results. Another systematic review of traditional Chinese medicine as monotherapy for epilepsy found no strong evidence for its effectiveness [38]. More high-quality evidence is needed before clinical recommendations can be made [39].

One study investigated a form of TCM known as Huazhuo Jiedu Shugan decoction (HJSD) in a rat model of epilepsy and concluded that HJSD is effective in reducing epileptic seizures [40]. Xiaocheng Yan’s clinical experience collection of epilepsy is a text compilation of clinical cases, in which the diagnosis and treatment of epilepsy using TCM were documented [41]. It is an important resource for TCM knowledge, and this medical record was analyzed. Commonly used Chinese medicinal herbs in the treatment of epilepsy were Ramulus Uncariae cum Uncis, Rhizoma Gastrodiae, Rhizoma Acori Tatarinowii, Bombyx Batryticatus, Radix Paeoniae Alba, and Cicada. These herbs were reported to be effective in the treatment of epilepsy, all without side effects. Shihogyejitang (SGT), a herbal medication, was studied in Korea, in which a retrospective study of pediatric patients with drug-resistant epilepsy using SGT were analyzed [42]. Over 44% of the patients showed a greater than 50% reduction in seizures, with 24% being seizure free. Side effects, such as a skin rash and fever, were reported.

*Paeonia officinalis (P. officinalis)* is a native perennial herb of Southern Europe and Western Asia used in traditional medicine with anticonvulsant effects. In an open-label study, 32 children with intractable seizures were given 70% hydroalcoholic extract of *P. officinalis* as an adjunct therapy [43]. After 4 weeks, there was a significant decrease in seizures, with 62.5% of the participants having a 50% or more decrease in seizures. This activity can be attributed to the presence of various bioactive components, such as flavonoids, polyphenols, and monoterpenes.

*Nigella sativa* (black cumin seed) is a traditional medicine known to have anticonvulsant effects. Most of *Nigella sativa*’s properties are associated with its extract thymoquinone. In Iran, black cumin seed is a commonly used natural remedy. One study conducted a double-blinded crossover clinical trial in which pediatric participants with refractory seizures received either the extract of black cumin seed or AEDs. The mean seizure frequency was reported to be significantly decreased during treatment with the extract [44].

A double-blind, placebo-controlled, randomized study showed that the administration of thymoquinone as an adjunct therapy significantly reduced seizures after 4 weeks. Additionally, 3 out of 22 patients were seizure free during the treatment period compared to none in the placebo group. The most common side effect was nausea, but the patients reported a better quality of life after the administration of thymoquinone [45].

Rhynchophylline (Rhy) is a plant highly used in traditional Chinese and Japanese medicine. In the study that analyzed the pharmacological mechanisms of Rhynchophylline, it was shown that Rhynchophylline is a promising compound for the future development of anticonvulsants due to its anti-inflammatory role. The study identified 20 candidate targets of Rhynchophylline against epilepsy that were important in renin secretion, morphine addiction, the neuroactive ligand–receptor interaction, and cGMP PKG signaling [46]. Rhy may be a promising compound in drug development for epilepsy, and further studies are needed.

1b.Ayurveda

Ayurveda, translated in Sanskrit as the “knowledge of life”, is an ancient alternative medical system originating from India around 5000 BC. Ayurveda takes a holistic approach to treating the person as it focuses on the balance between mind, body, and spirit. While stipulations exist about whether Ayurvedic medicine is pseudoscience, the research is ongoing as to its efficacy in treating epilepsy [47].

Epilepsy is denoted as “*Apasmara*” in Ayurveda, one of the *maharoga*, a group of eight diseases identified as causing serious illness. The causes of epilepsy—grouped as *Acharya Sushtra*—are believed to be due to the ingestion of unhygienic food, aggravation of body balance, and repression of natural body activities that affect mental health. In treating Apasmara, Ayurvedic medicine has identified approximately sixty herbs that have been shown to have antiepileptic effects, such as decreasing cholinesterase activity and inhibiting pentylenetetrazol (PZT)-induced seizures. While no empirical evidence exists, Ayurvedic medicine has been shown to have fewer side effects upon prolonged usage compared to epileptic drugs [48].

Among the sixty herbal medications that have been identified, two have been the focus of research—*Vacha* and *Brahmi*. *Vacha*, or *Acorus calamus*, is a medicinal plant with scented rhizomes that is known as a neuroprotective brain tonic. They can be found in the Yelagri Hills of Tamilnadu, Chennai, in November when they are fully matured. In experimental animal studies, Vacha has exhibited anticonvulsant activity, making it a prime candidate for epilepsy treatment. An experimental animal study was completed in which four groups of rats were administered either distilled water (control), Raw Vacha (RV), Shodita Vacha (SV—purified of toxins), or Phenytoin, a standard epileptic drug. Maximal Electric Shock (MES) was administered to all the rats to test the efficiency of the anticonvulsant activity of each drug. RV showed 31.76% protection against an MES-induced seizure while SV had 36.48%. In comparison, Phenytoin showed 91.74% protection. The study shows that Vacha, specifically purified Vacha, has anticonvulsant activity and has similar mechanisms to antiepileptic drugs through GABA-mediated synaptic inhibition [49].

*Brahmi*, or Bacopa monnieri Linn, is an Ayurvedic herb used for treating neuropsychiatric disorders. It can be found in wetlands and marshes in Sri Lanka, China, and Nepal. The herb is known to have active elements, such as alkaloids and bacosides, which contribute to its anticonvulsant mechanism. Rat studies show that Brahmi downregulates the NMDA R1 gene expression, reducing the frequency and severity of seizure onset. In addition, when Brahmi was administered to rats treated with Phenytoin, it reversed the cognitive impairment caused by the antiepileptic drug. While clinical studies are yet to be completed for Brahmi, it is frequently used for epileptic treatment in Uttarkhand, India [50].

While Ayurvedic medicine shows potential for anticonvulsant activity in epilepsy, its side effects need to be considered as well. There is the potential for an herb–drug interaction that can increase the risk of seizures. The proposed mechanism as previously mentioned is that herbal medications can alter cytochrome P450 enzymes and P-glycoproteins [15]. Another consideration is that Ayurveda incorporates metals such as lead, mercury, and arsenic as adjuvants as ancient texts indicate that metals enhance drug delivery in the body. Patients may develop acute lead toxicity, as seen in the case of a 26-year-old man who took herbal medications regularly for his epigastric pain [51]. Ayurveda is highly utilized in Asia, and more laboratory and clinical research is needed.

2.Supplements/Pills

In a study conducted in India, 108 patients taking CAM solicited by specialized alternative medicine clinics presented to an allopathic hospital with status epilepticus, uncontrolled seizures, or drug toxicity. Upon lab testing, conventional anti-seizure drugs (ASD) were detected in the patient’s serum indicating that the unlabeled CAM pills likely contained conventional anti-seizure drugs [9] as the patients were not prescribed ASDs; however, the presence of ASD in CAMs was not verified through further testing of the product. Phenytoin, carbamazepine, valproic acid, and phenobarbital were among the ASD drugs detected on the serum work-up and the levels ranged from subtherapeutic to supratherapeutic. The study found that the most probable reason for the patients’ loss of seizure control was the sudden withdrawal or overdose of ASDs content from the pills. It is unclear how an overdose of ASDs would cause seizures. This may have been mistakenly written.

Omega-3 supplementation has been studied in patients with epilepsy [52]. In a double-blinded, randomized control study, sixty children with ADHD and intractable epilepsy were enrolled and assigned to the omega-3 supplementation or placebo group in addition to antiepileptic drugs. The frequency and severity of seizures were assessed at baseline and monthly. Omega-3 supplementation was found to significantly decrease the monthly seizure frequency after 6 months compared to the baseline measurement before supplementation [53]. A meta-analysis on the use of omega-3 supplementation showed a beneficial effect on the seizure frequency in adults and children with epilepsy [54].

In an adult study on CAM use in patients with epilepsy, commonly used supplements were multivitamins and folic acid. Kaiboriboon et al. found herb and supplement problems used by the patients to have cytochrome p450 inhibition properties (St. John’s wort, garlic, grapefruit juice, folic acid, and Echinacea) [55]. Seven products contained ingredients reported to have the potential to induce seizures, such as gingko, ginseng, black cohosh, evening primrose, ephedra, dehydroepiandrosterone (DHEA), and caffeine [55]. While this study was conducted on adult patients, younger patients may also be at risk of using similar products.

A few studies have investigated the role of probiotics in treating epilepsy as it is believed that there is a gut microbiota role in epilepsy. A small prospective study found probiotics to be effective in improving quality of life [56].

Melatonin has been reported to be anecdotally helpful in improving seizure control, particularly myoclonic and nocturnal seizures. However, it is not clear whether this effect is through improved sleep quality [57].

3.Over-The-Counter Cannabis Products:

Cannabis products are increasingly used and sought after for children with refractory epilepsy in the United States with over-the-counter products increasingly growing in popularity. The pharmaceutical formulation of cannabidiol, “Epidiolex^®^”, is Food and Drug Administration (FDA) approved for the treatment of epilepsy in refractory Dravet syndrome and Lennox–Gastaut syndrome [14] and tuberous sclerosis [58,59]. The medical community and families of children with rare neurological disorders are also turning to pharmaceutical investigation and the development of CBD in hopes of finding new effective treatment. Other various cannabis products being used include medical marijuana and hemp oil. Cannabidiol (CBD) is widely available for purchase online or in stores and can be found in products of numerous forms, from gummies and drinks to shampoos and creams. The issue is that it is claimed to be CBD, but it is actually a mixture of several cannabis derivatives. As the interest in CBD products continues to rise, it is especially important to understand the current CBD research. Parents with children with refractory epilepsy are more likely to turn to CAM, particularly cannabis-related products, to attempt to control their child’s seizures [13]. Children with early-onset drug-resistant epilepsy and a higher seizure burden tend to suffer from greater neurodevelopmental problems. Therefore, research on the safety and efficacy of cannabidiol would be particularly useful for pediatric patients with epilepsy.

Historically, the Marijuana Tax Act passed by Congress in 1937 gave the Drug Enforcement Agency regulatory control over cannabis, criminalizing the possession of marijuana across the country [60]. Consequently, scientific, and medical research on cannabis products such as marijuana dwindled. Currently, state laws differ regarding marijuana use recreationally and/or medicinally, with 23 states taking action for legal use. While there is past and current clinical trial research investigating the therapeutic usage of CBD in the field of epilepsy, more high-quality clinical trials are needed to understand the safety and efficacy of CBD usage in patients with epilepsy that do not qualify for the use of Epidiolex.

3a.Pharmacology of Cannabinoids

The naturally grown plant cannabis contains over 80 different compounds, called cannabinoids, two of the most known and studied being delta-9-tetrahydrocannabinol (THC) and CBD [61,62]. THC is widely known for its psychoactive effects that create a feeling of being “high,” whereas CBD is non-psychotropic and therefore ideal for medicinal use and research. Research on THC and CBD is difficult as there is much variability in the plant product content and strains due to various growing techniques and processing. Furthermore, these products lack pharmaceutical quality control and other regulatory processes that ASMs are typically subjected to.

The three types of cannabinoids are phytocannabinoids (found naturally in cannabis plants), endocannabinoids, and synthetic cannabinoids. Endocannabinoids are chemicals found throughout the central and peripheral nervous system that activate cannabinoid-associated receptors. Synthetic cannabinoids are human-made and have been shown to be less effective with a higher risk of adverse events; therefore, natural cannabinoids are the main subject for research purposes.

3b.Mechanism of Cannabinoids

The cannabinoids’ pharmacological mechanism of effect primarily lies in the interaction with the endocannabinoid system consisting of cannabinoid receptors, such as the two identified receptors, CB1 and CB2. CBD modulates the enzyme metabolism of this system and plays a role in ion channel conductance. CBD, in contrast to tetrahydrocannabinol (THC), lacks psychoactivity and is therefore often more desirable for medicinal research.

CBD in low doses may impact the gut microbiome according to a study performed on a mouse model investigating the molecular effects of CBD [63].

3c.Data on the use of cannabis in children with epilepsy

The use of CBD for refractory seizures was popularized in mainstream media in 2010 when a girl named “Charlotte Figi” suffering from Dravet syndrome with a seizure burden of about 300 seizures per day began taking a CBD strain grown by the Stanley Brothers containing high CBD and low THC. After taking this CBD strain, her seizure burden dropped to 20 seizures a month. This story ignited a wave of parents also desperate to look for treatment for their child’s seizures. “Charlotte’s Web”™ is currently available for purchase online. There are numerous anecdotal reports for the use of CBD by parents for their child’s seizure. Parents administering oral CBD have reported improvements in mood, seizure reduction, and language and motor skills [64].

A cross-sectional study conducted in the U.S. interviewed parents about their use of cannabis for the treatment of their child’s epilepsy. Parents with the necessary financial means described testing their cannabis product for strength and contamination before beginning or when buying from a new supplier. This study highlighted that parents were often knowledgeable about CBD use and invested considerable time to inform themselves about CAM [65,66].

A nonrandomized controlled trial investigated the use of 4.2% topical CBD transdermal gel for the treatment of developmental and epileptic encephalopathies in children and found it to be well tolerated and have potential for seizure reduction [67].

4.Acupuncture

Acupuncture, an ancient practice that originated in China more than 3000 years ago, is an integral component of Traditional Chinese Medicine (TCM). Focused on the concepts of *Qi*, *Yin*, and *Yang*, acupuncture consists of thin needles being inserted at specific points on the body to relieve disease-specific symptoms.

Various clinical studies of the usage of acupuncture for epilepsy exhibit a reduction in seizure frequency, an improvement in electroencephalogram results, and the alleviation of the severity of epilepsy [68,69,70,71,72]. Acupuncture may act on the thalamus, which plays an integral role in epilepsy generation and propagation. In acupuncture, the “Dazhui” acupoint (GV14, the depression between the 7th cervical and 1st thoracic vertebra along the posterior midline) has been shown to inhibit pentylenetetrazol (PTZ)-induced epilepsy activity in the ventrobasal thalamic neurons [70].

While the efficacy of acupuncture is suggested by the results of several case studies of epilepsy, there is still no evidence from randomized controlled trials (RCTs) of its efficacy. To address this research gap, a study has been proposed in which 120 patients with post-stroke epilepsy ranging from the age of 18 to 75 years will randomly be assigned to either receive acupuncture or sham acupuncture alongside their routine treatment. The study will be measuring the adverse events and the number of seizure-free patients after the 8-week-long treatment course and is expected to start in December 2022 [73]. Adverse events reported to be associated with acupuncture are pneumothorax, central and/or peripheral nervous system injury, infections, syncope, organ and/or tissue injury, and hemorrhage [74].

5.Chiropractic care

Chiropractic care is a type of CAM that involves the diagnosis and treatment of musculoskeletal disorders, particularly the spine. Treatment involves the manipulation of the spine, joints, and/or soft tissues but also includes exercises and other lifestyle recommendations. Although there is currently insufficient scientific evidence to support chiropractic care in the treatment of pediatric epilepsy, there are reports of its benefit.

In a review of 17 case reports of pediatric epilepsy where patients received chiropractic care, vertebral subluxation correction resulted in a decrease in seizure activity [75]. This suggests that chiropractic care may be a potential non-pharmaceutical approach to pediatric epilepsy; however, further scientific studies are needed to investigate the effectiveness of chiropractic treatment in children with epilepsy. With spinal manipulation in chiropractic care, especially of the upper spine, the most common serious adverse events were due to vertebral artery dissections [76].

6.Religious and Cultural Practices

Although CAM for treating epilepsy is popular all over the world, its practices can vary regionally. It can be influenced by culture, history, religion, level of education, and individual interests. Records of the description of epilepsy and its treatments in the Middle East were discovered to date back to ancient times. The wealthy directory of CAM and medicinal plants, in particular, distinguishes it from other regions [77]. The literature has described the various patterns of use among the different countries in that region.

A study from North Jordan found that religious beliefs played a large role in parents supplementing conventional medicine with CAM for their child seen at a pediatric neurology clinic. Of the 176 patients, 99 parents (56%) used CAM for their children’s neurological illness, including epilepsy. Frequently used CAM options were prayer/reciting Quran (77%), religious healers (30%), olive oil massage (32%), and honey (29%) [10]. Religious beliefs (68%), fathers older than 30 years, and mothers with basic school education were the factors associated with CAM use. None reported a lack of trust in modern medicine as the reason behind resorting to CAM methods. It was noted that honey consumption was based on religious beliefs as the healing power of honey is highlighted in Islamic teachings.

One study investigated 304 epilepsy patients above the age of 18 about the usage of CAM in the management of their attacks. The parents of the patients usually resorted to praying (99.3%); keeping their children away from the effects of smoking (79.8%); feeding their children walnuts (79.6%), butter (59.2%), and bone marrow (58.6%); providing their children with good quality sleep (58.6%); and enabling them to play games (51%) [21].

Islam is the most common religious practice in the Middle East region. Prayers, blessings, or spiritual healing can aid in maintaining an improved sense of well-being [78]. Taking these cultural and religious factors into consideration, prayer is one of the most used approaches by parents in CAM. In Saudi, 79 mothers were questioned regarding the reasons they resorted to CAM use in their children with chronic conditions, including epilepsy. A total of 42% of the families used CAM for their children, and 57% of those had not asked for medical help beforehand. In addition, 82% relied on religious healing, and oral preparations or herbs were used in 30 % of the patients [79].

In Sudan, supernatural powers as a cause of epilepsy is a general belief [80]. Traditional healers are consulted for epilepsy treatment and are trusted. In a study that interviewed 180 parents of epilepsy patients, 58 (32.2%) interviewees believed epilepsy is due to supernatural causes. A large majority of them (70.5%) used CAM (e.g., incantations (45.6%), spitting cure (37.2%), and ritual incensing (36.7%)) to treat epilepsy [81].

Traditional Arab and Islamic medicine (TAIM) is common in the Middle East. Research on TAIM herbs has been carried out in many countries in the Middle East, such as Syria, Yemen, Egypt, Iran, and others [82]. More than 250 species from different families have been categorized in massive archives of plant species and are used by some to this day [77,83,84]. The study reviewed five of the most important Iranian herbal books between the 10th and 18th centuries, one of which was the Canon. The Canon, which means ‘‘The Law’’, was one of the most significant contributions made by the famous Persian physician Avicenna in the Western world (980–1037 AD). It is considered the biggest medical encyclopedia of its time [85]. *Aristolochia longa* L., *Bryonia dioica* Jacq., *Ferula persica* Willd., *Lavandula stoechas* L., and *Paeonia officinalis* L. were mentioned in all five books reviewed, demonstrating their significance through hundreds of years. “*Peganum harmala* L.” and “*Ruta graveolens* L.” are examples of known plants for the treatment of epilepsy in ancient Egypt [85]. “Mariam’s palm”, or *Anastatica hierochuntica* L. is used for the treatment of seizures by Bedouin people in the Sinai desert [86].

In a cross-sectional study of 101 patients in the Sultanate of Oman, 73.3% were using CAM for epilepsy. Over half of the CAM users believed their disease was due to spiritual possession, contemptuous envy, or sorcery. CAM use was also found to be linked to the following social factors: age of >30 years, low family income, having basic school education, and unemployment [87]. 

Additionally, decreased regional access to anticonvulsant drugs may drive individuals to resort to CAM [88]. For example, newer drugs such as Felbamate, Tiagabine, Zonisamide, Briveracetam, Rufinamide, and Stiripentol are not available; on the other hand, Lacosamide, Eslicarbazepine, and Perampanel are only available in the richer gulf countries of the region [12].

7.Yoga

Medical yoga has been shown to have health benefits and is used for treatment or adjunct therapy for many diseases. An increase in stress is associated with an increase in seizure frequency [84]. Yoga helps reduce stress and cortisol levels and restores the balance between the parasympathetic and sympathetic systems [89]. This can be attributed to “yoga breathing.” Because emotional states can increase respiratory demand, yoga helps to reverse this by increasing parasympathetic activity. Stretching the alveolar receptors, baroreceptors, and chemoreceptors in the respiratory system transmits information to the nervous systems that regulate emotion, behavior, cognition, and perception. Reducing stress is a method to also diminish the allosteric load. The allosteric load is the impact that deviation from homeostasis has on the body and disease progression, and when stressed, the allosteric load increases [89]. The practice of yoga has been shown to diminish the allosteric load through reducing stress reactivity and regulating the GABA release, the primary inhibitory neurotransmitter. Lower than normal levels of GABA have been associated with epilepsy, and yoga practices increase the levels of GABA along with serotonin and dopamine, which are protective from other psychiatric disorders (Streeter).

One prospective pilot study used yoga as an intervention for five children with generalized epilepsy, absence epilepsy, and/or complex partial epilepsy [90]. The subjects engaged in 2–3 home sessions over 6 months. The results showed that after a 6-week period with 2–3 sessions per week, there was a downward trend in anxiety and depression symptoms and no seizures were reported during the study. Another similar study was a randomized controlled trial involving 20 patients, showing that yoga is helpful in reducing seizures and improving EEG results after 6 months compared to the control group [91]. The limitations included a small sample size and the lack of a randomized controlled study.

Additionally, even though yoga requires a significant amount of time, it can be practiced for free and at home, which reduces barriers for patients to receive therapy [23].

8.Diets

Because nutritional factors play a significant role in neurological regulation, dietary intake is heavily studied in patients with epilepsy [92]. Apart from the ketogenic or Atkins diet and dietary considerations in children with metabolic syndromes, patients or families may turn to diet therapies that are self-imposed. The Mediterranean diet is high in plant-based foods and seafood and low in processed meats and sweets. A cross-sectional study looked at the nutritional status of the Mediterranean diet on 85 children with epilepsy. Adherence to the diet was measured with the Mediterranean Diet Quality Index (KIDMED). The results showed a positive statistical relationship between the KIDMED score and the levels of magnesium, carbohydrates, iron, potassium, calcium, and soluble fiber. The results showed that the average number of seizures per week in children with a moderate to high adherence to the diet was lower than the children with a low adherence; however, it was not statistically significant [93]. Children with a higher adherence to the diet may be more likely to also be adherent to ASDs, which may have contributed to the results. Additionally, the Mediterranean diet has numerous other positive health benefits and is potentially easier to adhere to than the ketogenic diet. Conversely, the diet is also high in sodium, which has been associated with hypertension and cardiovascular disease, and children were found to have a higher sodium intake when there was a stronger adherence to the diet.

Another study looked at the impacts of a gluten-free diet on patients with celiac disease and epilepsy. Studies report an association of celiac disease and epilepsy, and patients with refractory epilepsy may be tested for celiac disease depending on the clinical history [94]. After five months of a gluten-free diet, six of seven patients had their seizure activity successfully controlled and anticonvulsant medication was stopped. The limitations include but are not limited to the sample size.

Food allergies have also been linked to epilepsy, and an “elimination diet” is a diet that patients with refractory epilepsy turn to in this case [95]. It is an eating plan which removes foods believed to cause food intolerance. Studies have speculated that the inflammatory processes that elicit epileptic episodes are largely attributed to allergic reactions. In a prospective study of 34 patients aged 3 months to 16 years old with non-IgE-mediated and mixed food allergies based on skin and serum testing, an elimination diet was instituted over 12 weeks and the seizure frequency was measured following the intervention. There were six food eliminations, which included cow’s milk, egg, peanut, soy, wheat, and sea foods. After 12 weeks, it was found that 17 patients did not have any seizures after 8 weeks and 12 patients had a significant decrease (51–99%) in the number of seizures.

In a single-subject analysis of a 3.5 year old, the Feingold diet was introduced to reduce seizures [96]. The Feingold diet eliminates natural salicylates and artificial colors and flavors. The child was given the Feingold diet and then removed from it three times. Each time the diet was added, there was a notable reduction in the seizure frequency. The seizures were eliminated after one year. However, due to the Feingold diet decreasing the intake of carbohydrates and potentially some essential vitamins, it is recommended that this diet only be implemented under medical supervision. This is a single case report and hence the efficacy of the Feingold diet may need further research.

In a study conducted in Tehran, Iran, most of the parents of children with epilepsy did not believe in food playing a role in improving or worsening epilepsy. However, this study is limited to a survey of 155 families of a particular location [97].

9.Music Therapy

For the last 20 years, there have been numerous studies on the effects of listening to classical piano music by Mozart.

In a systematic review of eight studies, the Mozart Effect was shown to be a potential adjunctive treatment for children with epilepsy. This can be due to the music increasing dopamine, the parasympathetic tone, and sensorimotor circuits [98]. A 2014 pediatric RCT study showed that listening to Mozart K.448 reduced epileptiform discharges and seizure recurrence in children who had one unprovoked seizure of unknown etiology [99]. A long-term (up to 6 months) beneficial effect of listening to Mozart K.448 was found [99,100]. Larger randomized control studies are needed to determine the effectiveness. Further research would elucidate whether these beneficial effects exist only while listening to specific types of music or may persist after listening to the music.

10.Aromatherapy and Olfactory Stimulation or Training

Aromatherapy encompasses using fragrant essential oils as a treatment for various conditions through different methods of use, such as inhalation, massage, bathing, and perfume. Aromatherapy has been studied and tried as a form of behavioral therapy in people who experience seizures preceded by an aura. It is thought that inhaling the scent at the start of a seizure may help reduce the severity or chance of a seizure [101]. Yilmaz et al. studied the effect of olfactory training with lavender aroma in 24 patients with drug-resistant epilepsy [102]. The results demonstrated a decrease in seizure frequency and duration and an increase in quality of life that were statistically significant. In Isler et al.’s study conducted in Turkey, parents reported using various aromas to stop their child’s seizures, such as the smell of a lemon cologne, onions, and garlic [21]. In South East Asia, smelling shoes is a method to cease seizures [103].

#### 3.1.3. Perceived Effectiveness and Adverse Effects of CAM Therapies

There are many potential benefits of CAM, especially considering those generally known to be of low risk, including yoga and prayer. In the Beattie et al. study conducted in Alabama, 75% of those who prayed reported great benefit, and none reported side effects. Parents often noted that anti-seizure medication (ASM) side effects or the lack of efficacy were reasons that they initiate CAM for their child. The chronic use of anti-seizure medication may cause side effects such as behavioral changes and various systemic and metabolic disturbances, requiring vitamin supplementation. Among the various reasons parents turn to CAM, common reasons include the goal of improving their child’s seizure or to alleviate pharmacological medication side effects. While the perceived safety of CAM attracts people to it, some forms of CAM have the potential to do harm. Certain types of CAM have few or no interactions with pharmacological ASM, such as yoga and osteopathy; however, biological substances, such as herbal supplements, may interfere with ASM or have proconvulsant properties [15].

#### 3.1.4. Reasons for Using CAM

There are numerous reasons for using CAM, such as social, cultural, and personal motivations. Another factor to consider is the perception that CAM is safe [104]. In Beattie et al., 5% percent of the patients used CAM because they believed that prescribed medications were harmful, 14% because the prescribed medications were ineffective, 2% because CAM resonated with their beliefs, and 9% because CAM worked on others [19]. The majority reported that they were willing to try anything that would improve their child’s epilepsy.

CAM use was found to be positively correlated with the belief that conventional medicine is ineffective, costly, and/or is associated with side effects [105,106,107,108,109]. In a German study, the majority of participants were compliant with conventional care while using CAM [23] but were significantly less satisfied with conventional care than non-CAM users. Reasons patients cited for using CAM included a reduction in seizure frequency, the treatment of ASM side effects, or management of other health problems.

In studies conducted in Uganda, parents often reported the belief that their child’s seizure had a supernatural or spiritual cause, which led to the pursuit of CAM before seeking conventional evidence-based medications [110].

A survey study in the Republic of Guinea, a country in sub-Saharan Africa, confirmed the important influence of traditional healers on the community’s beliefs and perceptions about epilepsy [111]. Most patients with epilepsy visited traditional healers of more than 3 years prior to starting medical therapy and initiating medications regardless of age, gender, or household education level. This was due to the widespread, inaccurate beliefs about the cause and transmissibility of epilepsy. Collaboration with traditional healers is essential for public health education regarding epilepsy.

Table 2 summarizes the numerous reasons parents have for using CAM, predictors of CAM use, and whether the parents had discussed CAM usage with their child’s physician.

#### 3.1.5. Predictors of CAM Use

Factors that suggest increased seizure severity were often noted to be predictors of CAM. Zhu et al. reported that an increased number of annual seizures, a high number of anti-seizure medications taken daily, and the prior use of the ketogenic/Atkins diet or Epidiolex was associated with CAM use. Goker et al. reported male sex and the ineffectiveness of anti-seizure medications tried as potential CAM predictors. Kenney et al. also reported that patients who were less satisfied with their prescription medications were more likely to use CAM [20]. This may also reflect that these patients may have less tractable epilepsy. Past use of CAM was reported to be a reliable predictor of current CAM use [29].

The educational level of caregivers may influence the decision to use CAM. Although Doering et al. did not find an association between the caregiver’s education and their decision to use CAM, Chen et al. found that caregivers with high school education served as a predictor for CAM use, and Kenney et al. demonstrated a statistically significant association among caregivers’ spouses having at least some college education and the use of CAM for their child. Overall, Kenney et al. did not find a statistically significant association between families’ education level and CAM use [18].

In the Jordan study, factors significantly associated with CAM use were a speech delay of the child, parents’ belief in its usefulness, father’s age greater than 30 years, and mother’s education level less than high school [10]. Education level and its association with CAM use appears to differ in different geographical locations based on the literature. It was postulated that the association between CAM use and speech delay was likely explained by the lack of accessible speech therapy services in the community.

In the Alabama study, spiritual practices such as praying or holding a religious healing ceremony were found to be especially common among the respondents, with 88% of the respondents having prayed for their child’s epilepsy and 15% having held a religious ceremony to heal their child’s epilepsy [19]. Not surprisingly, a large percentage (85%) of the respondents identified as Christian.

The highest independent predictor of CAM use found in the German study by Hartmann et al. was the presence of adverse drug side effects from the anti-seizure medications that the child was on. Patients who experienced side effects were 5.6 times more likely to have used CAM compared with patients who did not experience any side effects.

#### 3.1.6. Patient–Physician Communication Regarding CAM Use

Considering the popular use of CAM in the world and the potential of many of the different types of CAMs to cause side effects or drug interactions, patient–physician discussions about the safe usage of CAM is important. J.F. Beatie et al. found that the majority of doctors or nurses did not inquire about CAM usage [19]. Furthermore, a large percentage of patients stated that they had not discussed using CAM with their child’s physician (Table 2). Liow et al. found that less than half of the participants indicated that their child’s physician was aware of their CAM usage [112]. Caregivers or parents may not voluntarily offer such information unless asked. Some patients did not believe sharing such information was necessary. Other potential reasons for the lack of voluntary disclosure of CAM use were hypothesized to include a fear of judgement, stigma, or rejection.

A survey conducted by Doering et al. at the University Children’s Hospital in Germany found that of those using CAM, only 53% had discussed this with their child’s neurologist (Table 2). Furthermore, over 85% of all parents wished to discuss CAM usage with their child’s neurologist but did not. Another survey of pediatric patients in a multiethnic community in Singapore concluded that caregivers used CAM for their child without thoroughly understanding the effects and drug interactions of CAM [27].

Healthcare providers routinely asking about CAM use can help extinguish hesitations by parents to discuss CAM. This will facilitate better education about the potential interactions by CAM on ASMs [22,112].

## 4. Discussion

The existing studies on the prevalence of CAM among the pediatric epilepsy population in the United States are limited to a small subset of geographic locations. The common types of CAM used among the pediatric epilepsy population, as reported in cross-sectional studies in different countries, include cannabis-related products, prayer, homeopathy, and multivitamins. We did not find high-quality clinical trials, meta-analyses, or systematic studies reporting adverse effects from these types of CAM modalities. CAM usage varies across different geographic locations; therefore, it is important for physicians to be aware of CAM usage among their patients. More high-quality studies would help predict what types of CAM are preferred in different regions.

Herb–drug interactions are unpredictable, especially considering that the quality and quantity of the active ingredients are not often specified and not regulated by the Food and Drug Administration [113]. While herbal medicines have the potential to cause severe side effects, only a few countries have laws requiring licensing for herbal remedies, such as Germany, France, Australia, and Sweden, to ensure the quality of herbal preparations. In the United States, products such as herbal medicines were removed from the Food and Drug Administration (FDA)’s jurisdiction due to the Dietary Supplement Health and Education Act of 1994. As herbs are considered dietary supplements, manufacturers are not required to demonstrate safety and efficacy before producing, selling, and marketing herbs. However, herbal supplements are still regulated by the FDA despite FDA approval not being needed to sell these products. As part of FDA regulation, herbal medicine falls under the dietary supplement category and must be free of contaminants, be accurately labeled, and refrain from making medical claims. Products not being regulated by the FDA does not mean there are no manufacturing standards in place as pharmaceutical-grade herbal products are commercially available as well as independent labs that test products for purity and contamination.

Additionally, as herbs may have variable interactions on the metabolism, absorption, and transport of ASMs, it is important for healthcare providers to ask parents about the usage of herbs, provide support, monitor for side effects, and encourage continued adherence to ASMs. Due to the lack of oversight by the FDA, parents should be encouraged to work with healthcare providers in educating themselves about the herb they would like to try for their child and thoroughly research the company to select a reputable manufacturer brand. Products should contain detailed labels including ingredients and potentially conduct independent testing of their products to ensure the purity and standardization of their products. However, even proper labeling does not ensure what is actually present in the product, and therefore careful monitoring is advised. Clinicians or parents/families can look up specific herbs on the National Center for Complementary and Integrative Health for evidence-based summaries of herbs [114].

Currently, Epidiolex remains the only pharmaceutical-grade CBD oil that is approved by the US Food and Drug Administration (FDA) for refractory epilepsy secondary to tuberous sclerosis, Dravet syndrome, and Lennox–Gastaut syndrome. Marijuana-derived CBD products that have not been approved by the FDA are made available in many states in the U.S., where medical and/or recreational cannabis is legal by state law [115]. While some states require a medical prescription to buy CBD products, elsewhere CBD products are sold only at licensed dispensaries. As a result of the 2018 farm bill (Agriculture Improvement Act of 2018, P.L. 115–334), CBD products derived from hemp are not subject to federal regulation and oversight. Therefore, hemp-derived CBD products are available for purchase over the counter.

Non-FDA-approved products have been reported to cause contamination with pesticides, causing illness and emergency room visits [116,117,118,119]. While pharmaceutical-grade cannabidiol (CBD) oil, also known as Epidiolex, has already been FDA approved for use, parents may purchase unregulated CBD oil, which may be unpurified, contaminated, or not contain CBD oil [120]. Furthermore, the inaccurate labeling of CBD products is a major concern for consumers. The FDA has tested CBD products on the market and issued several warning letters for the mislabeling of products [121,122]. This may lead to the inadvertent consumption of CBD products without consumers being aware or intoxication if dosing is not accurately labeled. Of note, THC intoxication has been shown to affect white matter development in the brain and impact cognitive functioning in adolescents [123].

As the popularity of cannabis-related products such as CBD continues to rise, it is important to consider how CBD products are obtained. Commonly, CBD products are sold online or in retail stores and are not regulated in terms of quality and quantity as rigorous testing is not required. One study evaluated CBD products purchased online and found that 26 percent of the products contained less than the listed labeled quantity [122]. It may be difficult for parents and patients to find healthcare providers knowledgeable about CBD products; however, healthcare providers can still play an instrumental role in guiding and supporting patients and their families in their pursuit. Certain companies provide third-party testing and a “Certificate of analysis (CAO)” for their CBD products, which provides information about the quantity of CBD and the purity of the product. Furthermore, regarding the use of CBD products, patients should be advised to start with a low dose and gradually increase to minimize side effects and overdosing. While parents often perceive CBD as a suitable and safe option for their child, side effects to be aware of include somnolence, decreased appetite, diarrhea, fatigue, nausea, and headache [124]. The positive side effects reported include improved cognition, sleep, and mood in addition to improved seizure control [125]. To minimize adverse effects, purchasing from a reputable brand with third-party testing and a CAO, ensuring the storage of the CBD out of reach of children, and starting with a low dose while gradually increasing may be helpful [126].

As cannabis-related products, particularly CBD, increase in popularity and more patients and their families seek CBD for therapeutic purposes, more clinical research is needed to delineate the safety and efficacy of CBD usage. The limitation in terms of research advancement in this area is due to restrictive and legal constraints surrounding the usage of cannabis products. This makes it difficult to obtain study approval from the Food and Drug Administration (FDA), the National Institute of Health (NIH), the research state’s department of health, and the institutional review board (IRB) standpoint, which can often take 1 or more years. A question that is often raised is whether there is a placebo effect that may be playing a role in some of the reported study outcomes. To study this, randomized, double-blind control trials would be the ideal way to elucidate whether this effect exists to any degree. Another technique would be to examine patients with electroencephalograms to determine whether there are electrographic changes associated with taking CBD over time. Other important research considerations are the impact of cannabis products such as CBD on areas that impact quality of life, such as sleep and social, behavioral, and cognitive functions. The incorporation of a full sleep study can also be used in clinical studies to help study CBD’s impact on sleep. Studies looking at the dosage, timing, and co-usage of CBD with anti-seizure medications are also critical for understanding.

Homeopathic drugs are not well studied and may have unknown and potentially hazardous effects. Cannabis-based products can be associated with gastrointestinal adverse effects, and dietary therapies such as the ketogenic diet can be associated with many adverse side effects. It is important for physicians to address these gaps in the research and potential substance interactions when advising patients on safe treatments.

Moreover, because most patients do not disclose using CAM to their physicians, it is important that active inquiry on the part of the physician or healthcare provider should take place to prevent unwanted side effects or drug interactions. CAM usage can also be documented for the optimal long-term care of the patient. Although data are limited, Beattie et al. reported inadequate parent–physician discussions regarding CAM in Alabama, USA [19].

As care for children with epilepsy at most institutions typically involves a multidisciplinary team, nurses, pharmacists, and other healthcare providers may also play a role in identifying CAM users, providing education, and communicating with the physician. An increased emphasis on a multidisciplinary approach should be encouraged to address the inadequate disclosure of CAM usage. Low socioeconomic status played a role in opting for CAM before seeking evidence-based care. Therefore, it is stressed that increased initiatives to educate the parents and families of the neurobiological causes of seizures and the effectiveness of anti-seizure medication may be beneficial to prevent a delay in care. A collaboration between spiritual healers and biomedical providers is another route that is being explored.

Further research on CAM prevalence among pediatric patients with epilepsy among specific geographic locations in the United States will be beneficial for physicians practicing in diverse locations with unique cultural and demographic differences. It will provide physicians with a better understanding of which patients are most likely to use CAMs and in what form, thereby improving the quality of their patient communication and education. Furthermore, all forms of CAM pose the danger of medication non-adherence to anti-seizure medications, risking seizure aggravation potentially associated with unintended consequences, such as sudden unexpected death in epilepsy, status epilepticus, accidental injuries such as drowning or falls, and/or developmental regressions. Routinely asking about CAM use by healthcare providers can help extinguish hesitations by parents to discuss CAM. This will facilitate better education about potential interactions by CAM on ASMs.

### Limitations

In general, CAM encompasses many different therapies and there is no clear consensus on what is considered CAM, making the study of the use of CAM difficult to compare across different studies. Of the various studies mentioned, the CAM definitions were different. Prayer, for example, is defined as a type of CAM in most U.S. studies, whereas European studies on CAM usually do not include prayer as a CAM practice. In Asadi-pooya et al.’s study, the ethics committee did not permit questions related to prayer/spirituality in the survey administered to participants [25]. The questions and options listed in surveys administered may additionally impact the results. Therefore, the comparability of the percentage of CAM users is limited and is a major limitation in comparing the prevalence of CAM use.

Furthermore, the method of data collection in the studies mentioned differed and likely played a role in limiting the comparability of CAM usage. For instance, Doering et al. sent questionnaires to the parents of patients, receiving a response rate of 44.7%. In contrast, Hartmann et al. approached patients during outpatient appointments or hospital stays, with a participation rate of 99%. Studies with lower response rates are more likely to skew the data regarding CAM use, leading to higher calculated usage rates.

Moreover, there may be many other relevant articles that may touch on CAM use in children with epilepsy that were not included.

## 5. Conclusions

Conventional treatments/therapies may be ineffective in some patients with epilepsy and can cause side effects. There are many cultural and regional influences on the patterns of CAM use. Further studies on the efficacy of the different types of CAM used, as well as the potential side effects and drug interactions are needed. The increased utilization of CAM in the United States necessitates that practitioners be aware of the various CAM modalities used by pediatric patients with epilepsy, so that they can facilitate the safe use of these therapies and provide holistic care. Caregivers or patients may not be fully aware of the effects of CAM and the potential side effects. An active inquiry about CAM usage on the part of providers is essential to this.

## Figures and Tables

**Figure 1 children-10-00713-f001:**
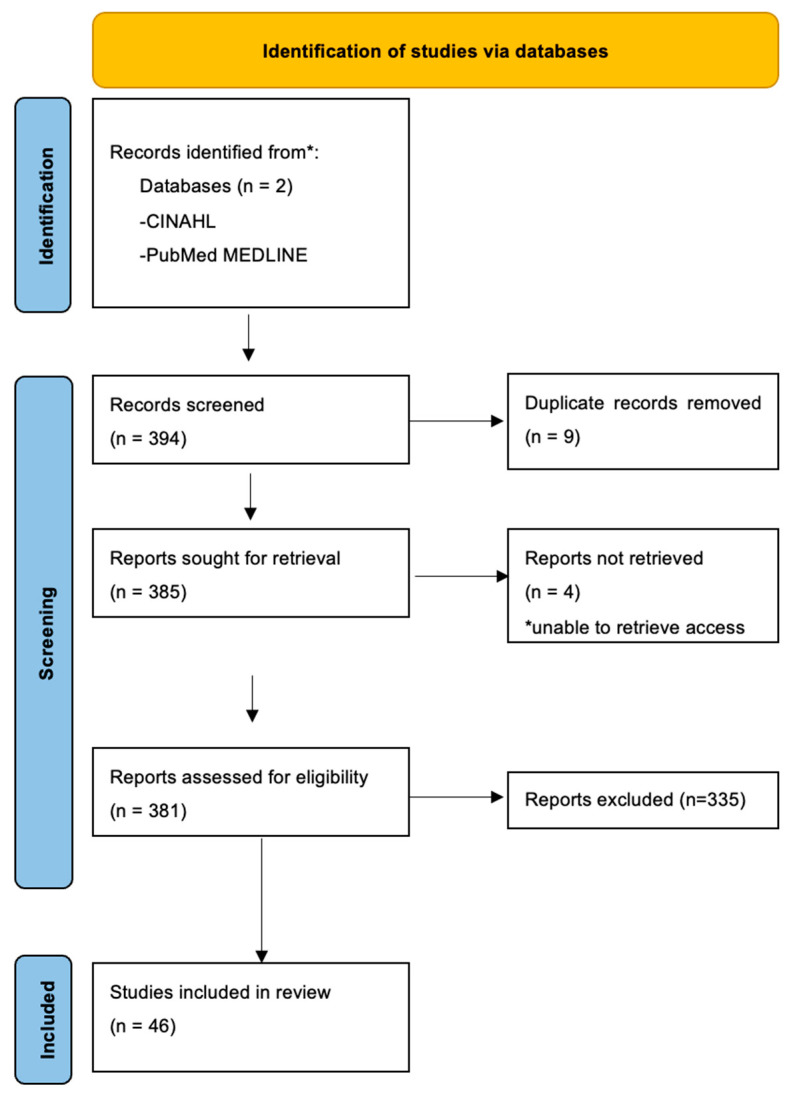
Search of Literature on CAM use in pediatric populations with Epilepsy. * Articles were excluded if access was unavailable through our institutions.

**Table 1 children-10-00713-t001:** Cross-sectional studies on CAM use among children with epilepsy.

	Location	% of CAM Use (*n* = # of Total Patients)	Common Types of CAM Used (% of Users)
Kenney et al. [18]	Minnesota, USA	35% (*n* = 107)	n/a
Beattie et al. [19]	Alabama, USA	26% (*n* = 225)	Massage, vitamin/supplements (5%) High prevalence of prayer was also noted
Zhu et al. [13]	Pennsylvania, USA	13% (*n* = 200)	Cannabis-related products (60%), chiropractic care (16%), aromatherapy (12%), homeopathy (12%)
Goker et al. [20]	Ankara, Turkey	22.6% (*n* = 69)	Prayer (91.3%)
Isler et al. [21]	Antalya Province, Turkey	100% (*n* = 304)	Prayer (99.3%), walnuts (79.6%), butter (59.2%), bone marrow (58.6%)
Hartmann et al. [22]	Leipzig, Germany	13% (*n* = 164)	Homeopathy (67%), osteopathy (57%)
Doering et al. [23]	Heidelberg, Germany	37.1% (*n* = 297)	Homeopathy (55.1%), osteopathy (24.5%), kinesiology (16.3%)
Jeong et al. [24]	Yangsan, Korea	21.9% (*n* = 389)	n/a
Tonekaboni et al. [25]	Tehran, Iran	44% (*n* = 133)	Prayer and wearing amulets (52.5%), dietary therapy (15%), herbal remedies (15%)
Asadi-pooya et al. [26]	Shiraz, Iran	10% (*n* = 98)	Herbal and traditional remedies (6%)
Chen et al. [27]	Singapore	27.5% (*n* = 178)	Multivitamins (44.4%), traditional herbs (42.4%), acupuncture (17.4%)
Lagunju et al. [28]	Ibadan, Nigeria	56.6% (*n* = 175)	Herbal products (39%), spiritual healing/prayer (34%), scarification (17%)
Gross-Tsur et al. [29]	Jerusalem, Israel	32% (*n* = 115)	n/a

**Table 2 children-10-00713-t002:** Reasons for using CAM, predictors of CAM use, and parent–physician discussions regarding CAM use.

	Common Reasons for Using CAM	Predictors of CAM Use	Parent–Physician Discussion Regarding CAM
Beattie et al. [19]	Conventional medicines were harmful, medication inefficacy	n/a	19% discussed using CAM with their child’s physician
Zhu et al. [13]	Dissatisfaction with the efficacy and/or side effect of conventional medicine	Prior use of Epidiolex and/or the ketogenic/Atkins diet, annual seizure frequency, number of anti-seizure medication currently used	80% of CAM users discussed CAM with their child’s physician
Goker et al. [20]	Belief in fighting the illness (spiritual), treat or cure epilepsy, dissatisfaction with current treatment	Male sex, resistance to antiepileptics	n/a
Isler et al. [21]	Reduce seizure frequency, belief that CAM is harmless	n/a	n/a
Hartmann et al. [22]	More natural, less side effects, desire to try everything	Adverse effects of anti-seizure drugs, higher number of seizures in the past year, lack of efficacy of conventional drugs, use of CAM by parents	76% discussed using CAM with their child’s physician
Doering et al. [23]	Reduce seizure frequency, treat side effects of anti-seizure drugs	Use of CAM by parents, parents who value a holistic and natural approach, longer disease course	53% discussed using CAM with their child’s physician
Tonekaboni et al. [25]	Recommended by relatives, medication inefficacy	n/a	16.7% discussed using CAM with their child’s physician
Asadi-pooya et al. [26]	Dissatisfaction with the efficacy and/or side effect of conventional medicine		
Chen et al. [27]	More natural, fewer side effects, treat side effects of anti-seizure drugs, stop seizures	Caregivers with secondary school education	75% reported that the healthcare provider did not ask them about CAM use
Lagunju et al. [28]	n/a	Social class of family and mother’s level of education (inversely related to CAM use)	Leading reason for the lack of CAM use was that the physician did not ask; only 30% of CAM users volunteered this information
Gross-Tsur et al. [29]	n/a	Past use of CAM, mother/father’s level of education higher, duration of disease, dissatisfaction with AED treatment	n/a

## Data Availability

Data used to support findings of this study are included in the article.

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
