# Peer review of "Use of Integrative, Complementary, and Alternative Medicine in Children with Epilepsy: A Global Scoping Review"

_children, 2023, doi:10.3390/children10040713_

Round 1

Reviewer 1 Report

The Materials and Methods section is quite brief. It would be helpful if the authors could provide more details about how the review was carried out, so that others can replicate their results (if they so choose).

Some materials and methods questions:

- Did the authors register the protocol with PROSPERO?

- Did the authors try including other search terms such as the acronym "CAM" to see if additional articles emerged? What about related terms such as "integrative medicine" and "supplemental medicine"? 

- Please be more specific with the Boolean search terms. Was the Boolean structured as: "complementary and alternative medicine" AND "epilepsy" AND "children" OR "pediatrics"

- Was an age limit used in selecting studies about children? If up to age 18, did the authors include "adolescen*" as one of the search terms? 

- Why was the time frame of the search limited to between January 2002 - December 2022? Why not search backward with a unlimited time frame? 

- what were the specific inclusion/exclusion criteria for the articles discovered by the initial Boolean search?

- Please explain why, "Following a review of the nine selected articles, a further literature review was conducted."

- Please provide more details about how you "searched for CAM use in geographical locations such as Africa, America (North/South), Middle Eastern and Asia."

- Please indicate how many articles were "excluded articles not specifically focused on the pediatric epilepsy population."

- Please provide a clearer flowchart of the selection of articles across the two search actions. It appears that 9 articles were identified in the first search, but how many articles were identified in the second search? Ten studies were included and discussed in the results section, but how many studies were excluded across the 2 searches and why were they excluded? 

Other Questions/Comments:

Line 298: "The Middle east region is mostly a Muslim nation where...."  (The Middle east region is not a nation)

Line 308: "...holds a prominent place in the middle (East??) region."

Line 323: "An important point worth discussing is how decreased access to anticonvulsant drugs in the region can push individuals to resort to CAM." Do you have evidence that this decreased access can push people to resort to CAM?

Line 599: "Most of the studies involving the pediatric epilepsy population used CAM in addition to conventional medicine." Given that this study looked only at articles of epilepsy and CAM, is this an accurate statement? It appears that the authors are stating that most studies of pediatric epilepsy that have been published used CAM, but that is not correct. 

Reviewer 2 Report

Zhu et al. reviewed the use of integrative, complementary, and alternative medicine in Children with Epilepsy. This review article is very informative and written well.

I have some minor concerns which need to be addressed before publication.

1.      The authors should expand the introduction using clinical data related to epilepsy in children.

2.      The authors have discussed the role of several natural products in the therapy of epilepsy in children, and it would be better if they discuss active ingredients of these natural products.

3.      The conclusion part should be revised.
